# Pharmacokinetic, Pharmacokinetic/Pharmacodynamic, and Safety Investigations of Lefamulin in Healthy Chinese Subjects

**DOI:** 10.3390/antibiotics12091391

**Published:** 2023-08-31

**Authors:** Yingying Hu, Qiong Wei, Xingchen Bian, Xinyi Yang, Jicheng Yu, Jingjing Wang, Haijing Yang, Guoying Cao, Xiaojie Wu, Jing Zhang

**Affiliations:** 1Phase I Clinical Research Center, Huashan Hospital, Fudan University, Shanghai 200040, Chinabianxingchen@huashan.org.cn (X.B.); yujicheng@fudan.edu.cn (J.Y.); wangjingjing@huashan.org.cn (J.W.);; 2National Clinical Research Center for Aging and Medicine, Huashan Hospital, Fudan University, Shanghai 200040, China; 3Research Ward of Huashan Hospital, Fudan University, Shanghai 200040, China; 4Institute of Antibiotics, Huashan Hospital, Fudan University, Shanghai 200040, China; 5Key Laboratory of Clinical Pharmacology of Antibiotics, Shanghai 200040, China

**Keywords:** lefamulin, healthy subjects, pharmacokinetics, safety

## Abstract

This study aimed to explore the pharmacokinetics (PK) and safety of oral (PO) and intravenous (IV) lefamulin in healthy Chinese subjects and to evaluate the efficacy of the intravenous administration regimen using pharmacokinetic/pharmacodynamic (PK/PD) analysis. This study was a randomized, open-label, single- and multiple-dose, intravenous and oral administration study. PK parameters were calculated, and the probability of target attainment (PTA) and the cumulative fraction of response (CFR) after IV administration of lefamulin 150 mg 1 h q12 h were analyzed with Monte Carlo simulations. Lefamulin exhibited extensive distribution. The mean steady-state AUC_0–24 h_ of 150 mg lefamulin IV and 600 mg lefamulin PO were 10.03 and 13.96 μg·h/mL, respectively. For *Streptococcus pneumoniae* and *Staphylococcus aureus*, based on the free-drug AUC over MIC ratio (*f*AUC/MIC) target of 1-log_10_ cfu reduction, the PK/PD breakpoints were 0.25 and 0.125 mg/L, respectively. The CFR was over 90% for both types of strains with 95% protein binding rate, suggesting that the regimen was microbiologically effective. Lefamulin was safe and well-tolerated. The PK of lefamulin in healthy Chinese subjects were consistent with that in foreign countries. Lefamulin demonstrated the microbiological effectiveness against *Streptococcus pneumoniae* and *Staphylococcus aureus*.

## 1. Introduction

Lefamulin (formerly BC-3781) is the first pleuromutilin antibiotic used for the systemic treatment of bacterial infections in humans. The pleuromutilin core and C-14 side chain of lefamulin can bind to the A and P sites of the 23S rRNA peptidyl transferase center (PTC) in the 50S bacterial ribosomal subunit through hydrogen bonds, hydrophobic bonds, conformation changes, and other ways, inhibiting bacterial protein synthesis by preventing the correct positioning of tRNA CCA and affecting the peptidyl transfer. The unique two-site binding mechanism of pleuromutilin selectively inhibits bacterial protein synthesis and reduces the risk of cross-resistance with other antibacterial agents. The binding sites of lefamulin in the highly conserved core of PTC result in strong binding and low rates of resistance development [1].

In vitro studies have shown that lefamulin exhibits broad-spectrum antibacterial activity against Gram-positive and Gram-negative aerobic and anaerobic bacteria, as well as against atypical bacteria, including drug-resistant strains such as methicillin-resistant *Staphylococcus aureus* (MRSA), multidrug-resistant *Streptococcus pneumoniae*, and macrolide-resistant *Mycoplasma pneumoniae* [2,3,4]. It demonstrates strong activity against respiratory pathogens. Studies have illustrated that the in vitro activity of lefamulin against Chinese clinical isolates is similar to that against isolates from Europe and America [3,4,5].

Lefamulin is available in tablet and intravenous (IV) formulations, and it is rapidly absorbed orally with an oral bioavailability of 25%. A 600 mg oral dose of lefamulin in a fasted state provides exposure similar to that of 150 mg IV infusion, supporting sequential oral administration after IV infusions. Multiple phase I clinical studies, one phase II clinical study in patients with acute bacterial skin and skin structure infections (ABSSSI) [6], and two pivotal phase III clinical studies in patients with moderate-severe community-acquired bacterial pneumonia (CABP) have been completed in the United States and other countries [7,8]. Lefamulin was approved by the United States Food and Drug Administration (FDA) in 2019 and the European Medicines Agency (EMA) in 2020 for the treatment of CABP based on the completed clinical studies [9,10].

However, the pharmacokinetics (PK) and safety of lefamulin have not been investigated in healthy Chinese participants (both oral and IV infusion). It is important to conduct pharmacokinetic/pharmacodynamic (PK/PD) analysis based on the PK of lefamulin in healthy Chinese subjects and the distribution of minimum inhibitory concentration (MIC) of common pathogenic bacteria causing CABP. This analysis is crucial for the rational use of lefamulin in Chinese patients with CABP.

## 2. Results

### 2.1. Demographic Characteristics

The study included 20 subjects aged 30 ± 5.1 years. Among them, fourteen subjects (70%) were male and six (30%) were female. Seventeen (85%) subjects were Han, and three (15%) were from ethnic minorities. The subjects had average height, weight, and BMI of 164.3 ± 7.06 cm, 61.6 ± 6.41 kg, and 22.8 ± 1.95 (kg/m^2^), respectively.

### 2.2. PK Characteristics

The PK parameters of lefamulin and its main metabolite BC-8041 for single- and multiple-dose IV and oral administration are presented in Table 1 and Table 2. The concentration-time curves of lefamulin and its main metabolite BC-8041 are shown in Figure 1 and Figure 2.

After administering 150 mg of lefamulin intravenously, it reached its peak concentration at the end of infusion and exhibited extensive distribution, with apparent volume of distribution (V_z_) of 332.7 ± 100.0 L. The apparent clearance (CL) was 23.5 ± 7.2 L/h, and the elimination half-life (t_1/2_) was 9.90 ± 1.2 h (Table 1). The t_1/2_ of the metabolite BC-8041 was 11.59 ± 2.89 h, similar to lefamulin (Table 2). The exposure ratio (AUC_0–∞_) of BC-8041 to the original drug was 4.3%.

Following single-dose oral administration, lefamulin was rapidly absorbed with time to peak concentration (T_max_) of approximately 1.38 h. The t_1/2_ of oral administration was 10.15 ± 1.70 h, which was not significantly different from IV administration. The area under the concentration-time curve (AUC) from 0 to infinity (AUC_0–inf_) after a single dose of lefamulin 600 mg orally was similar to that after a single IV dose of lefamulin 150 mg (7.71 ± 2.82 μg·h/mL vs. 6.90 ± 1.84 μg·h/mL) (Table 1). The T_max_ of BC-8041 was approximately 1.5 h, which was close to that of lefamulin. The t_1/2_ of BC-8041 was 9.72 ± 1.43 h, also similar to lefamulin (Table 2).

After five days of multiple-dose IV administration of lefamulin 150 mg every 12 h, the concentration of the drug had reached steady state. The peak concentration (C_max_) at steady state (C_max,ss_) was similar to that after single-dose, and accumulation ratio (Rac) based on C_max_ (R_ac(Cmax)_) was 1.0 ± 0.1. AUC accumulated slightly at steady state, and Rac based on AUC (R_ac(AUC)_) was 1.4 ± 0.1. The CL at steady state (CL_ss_) was 20.5 ± 5.9 L/h, and the t_1/2_ at steady state (t_1/2,ss_) was 12.7 ± 1.6 h, similar to those after single IV administration (Table 1). The R_ac(AUC)_ and R_ac(Cmax)_ of BC-8041 were 2.6 ± 0.6 and 1.8 ± 0.4, respectively. Based on AUC within dosing interval at steady state (AUC_0−tau,ss_), the exposure ratio (AUC_0−tau,ss_) of BC-8041 to the original drug was 5.45% (Table 2).

After five days of multiple-dose oral administration of lefamulin 600 mg every 12 h, the concentration of the drug had reached steady state. The C_max_ and AUC at steady state were slightly higher than those after a single dose, with R_ac(AUC)_ and R_ac(Cmax)_ values of 1.6 ± 0.3 and 1.3 ± 0.1, respectively. The AUC_0−tau,ss_ was 10.64 ± 3.23 μg·h/mL after 600 mg oral administration, slightly higher than that after 150 mg IV administration (7.84 ± 2.07 μg·h/mL). The degree fluctuation (DF) of oral administration was 178.4 ± 27.9%, significantly lower than that after IV administration (363.6 ± 32.8%) (Table 1). The R_ac(AUC)_ and R_ac(Cmax)_ of BC-8041 were 1.66 ± 0.36 and 1.14 ± 0.21, respectively. The exposure ratio of BC-8041 to the original drug was 26.4% based on AUC_0−tau,ss_. The mean dose-normalized absolute bioavailability of single-dose 600 mg oral administration of lefamulin was 28% (95% confidence interval: 24.3–31.7%) based on the PK parameter set (Table 2).

### 2.3. PK/PD Analysis

The probability of target attainment (PTA) of lefamulin 150 mg IV 1-h infusion for *Streptococcus pneumoniae* and *Staphylococcus aureus* as shown in Table 3 and Figure 3. The results showed that different protein binding rates significantly affected the PK/PD breakpoints. For *Streptococcus pneumoniae*, based on the free-drug AUC to MIC ratio (*f*AUC/MIC) target of 1-log_10_ cfu reduction, the regimen could cover an MIC_90_ of 0.125 mg/L when the protein binding rate was lower than 90%. The PK/PD breakpoint of lefamulin against *Streptococcus pneumoniae* was 0.25 mg/L, with a protein binding rate of 85% and an *f*AUC/MIC target of 1-log_10_ cfu reduction. For *Staphylococcus aureus*, based on the *f*AUC/MIC target of 1-log_10_ cfu reduction, the regimen could cover an MIC_90_ of 0.06 mg/L when the protein binding rate was lower than 90%. The PK/PD breakpoint of lefamulin against *Staphylococcus aureus* was 0.125 mg/L, with a protein binding rate of 85% and an *f*AUC/MIC target of 1-log_10_ cfu reduction.

When the protein binding rate was lower than 95%, the cumulative fraction of response (CFR) of lefamulin 150 mg IV infusion for 1 h against *Streptococcus pneumoniae* and *Staphylococcus aureus* were both more than 90%, with *f*AUC/MIC targets of 1-log_10_ cfu reduction (Table 4). Therefore, with a protein binding rate of 95% and a *f*AUC/MIC target of 1-log_10_ cfu reduction, lefamulin 150 mg IV infusion for 1 h could achieve the expected antimicrobial efficacy against *Streptococcus pneumoniae* and *Staphylococcus aureus*.

### 2.4. Safety Evaluation

Table 5 shows the treatment-emergent adverse events (TEAEs) after lefamulin administration in 20 subjects. Eighteen (90%) subjects reported lefamulin-related TEAEs, which were mild and moderate in severity. No severe TEAEs occurred, and there were no TEAEs leading to drug discontinuation, withdrawal from the study, or death.

After IV administration of lefamulin, the most common treatment-related TEAEs were infusion site reactions (pain, pruritus, erythema, etc.). The TEAEs associated with oral administration were mainly gastrointestinal disorders (nausea, abdominal discomfort, abdominal pain, etc.). No abnormalities were observed with laboratory tests accompanied by clinical symptoms or requiring treatment, vital signs, or ECG tests. Oral and IV administrations of lefamulin were well-tolerated and safe in Chinese healthy subjects.

## 3. Discussion

Community-acquired bacterial pneumonia (CABP) is a significant cause of morbidity and mortality worldwide that requires systemic antimicrobial treatment. Most common isolated pathogens associated with adult CABP include *Streptococcus pneumoniae*, *Haemophilus influenzae*, *Mycoplasma pneumoniae*, *Chlamydophila pneumoniae*, and *Staphylococcus aureus* [5,11]. Standard empiric antimicrobial treatments for CABP include β-lactams, β-lactam/β-lactamase inhibitors, doxycycline, minocycline, macrolides, and respiratory quinolones. However, in recent years, the bacterial resistance rates of *Streptococcus pneumoniae*, *Haemophilus influenzae*, *Staphylococcus aureus*, and *Mycoplasma pneumoniae* are increasing, making the empiric treatment of CABP more complex. Data from the 2022 CHINET [12] show that the resistance rate of *Streptococcus pneumoniae* to erythromycin is as high as 90%, and the resistance rate of *Haemophilus influenzae* to ampicillin is 70%. MRSA shows a 100% resistance rate to penicillin and benzoxicillin and a 73% resistance rate to erythromycin. Therefore, there is an urgent need to develop new antibiotics with unique mechanisms of action to prevent cross-resistance and provide new clinical treatment options for CABP, while further regulating the rational use of existing antibiotics to curb antibiotic resistance. Lefamulin is the first semisynthetic pleuromutilin antibiotic used systemically with unique mechanism of action and limited cross resistance to other antibacterial classes. Previous preclinical and clinical studies have confirmed the safety and efficacy of lefamulin. This is the first study evaluating the safety and PK profile of lefamulin following intravenous and oral administrations at single and multiple doses in healthy Chinese subjects and support clinical application of lefamulin in Chinese population from PK/PD perspective.

This study demonstrated that lefamulin was rapidly absorbed after oral administration, with median t_max_ values of 1.38 h and 1.75 h for single and multiple-dose administration, respectively. Approximately half of the subjects who received oral administration of lefamulin showed a bimodal curve, with the first peak occurring 20 to 45 min after administration and the second, higher peak (C_max_) occurring 1 to 2 h after administration, suggesting a mixed absorption process. There was no significant difference in t_1/2_ after IV and oral administrations, suggesting that the same dosing interval could be used for both routes of administration. Lefamulin was widely distributed, with a V_z_ of up to 300 L, suggesting extensive tissue distribution. Previous studies on lung tissue penetration showed that the ratio of total drug exposure in alveolar epithelial lining fluid to free drug exposure in plasma was approximately 5:1, and this high exposure in the lung tissues also highlights its potential in lung infection treatment [13]. Lefamulin is primarily metabolized by CYP3A4 and the metabolite BC-8041 is the main component that displayed similar elimination with lefamulin. The metabolite rates were estimated to be 4% and 25% for intravenous and oral formulation, respectively. A steady state was reached 5 days after multiple-dose lefamulin administration, both IV or oral, with similar exposure and no significant accumulation, supporting the sequential administration of lefamulin at a dose of 150 mg intravenously and 600 mg orally. As shown in Figure 4, the exposure of lefamulin in healthy Chinese subjects was comparable to that in Western populations [14,15], regardless of single- or multiple-dose administrations, and the PK characteristics after IV and oral administration were consistent with previous studies. As for CABP patients, the exposure of lefamulin in CABP patients was higher than that in healthy Chinese subjects. The C_max_ of lefamulin in CABP patients was about 1.5 times of that in healthy Chinese subjects and the AUC_0–24 h_ was about 4.4 times, regardless of IV or PO administration [16].

Considering the concentration-dependence of the protein binding rate of lefamulin, different protein binding rates were selected for PK/PD analysis in this study to evaluate their effects on lefamulin PK/PD breakpoint, PTA, and CFR. The results indicated that when the protein binding rate was lower than 90%, the MIC_90_ of *Streptococcus pneumoniae* and *Staphylococcus aureus* could be covered effectively at the administration regimen (150 mg IV infusion 1 h), resulting in a microbiologically effective response. When the protein binding rate was lower than 95%, the CFR could reach more than 90%. These results suggest that to evaluate the efficacy of the administration regimen, the effects of protein binding rates should be considered in PK/PD analysis of nonlinear protein-bound drugs, combined with the MIC distribution of bacteria. In the future, lefamulin exposure and its protein binding rate in epithelial lining fluid should be considered to further evaluate its efficacy to CAP.

## 4. Materials and Methods

### 4.1. Study Designs 

This single-center, randomized, open-label, two-period crossover study aimed to evaluate the PK, safety, and tolerability after single- and multiple-dose IV formulation (150 mg) and immediate-release oral tablets (600 mg) of lefamulin in healthy Chinese subjects. The study protocol was approved by the ethics committee of Huashan Hospital, Fudan University (approval number: 2019-374), and the trial is registered with Chinadrugtrials.org.cn (identifier: CTR20191230). The study design is shown in Figure 5.

### 4.2. Subjects

A total of 20 healthy women and men aged 18 to 45 years, with a body mass index (BMI) ranging from 19 to 26 kg/m^2^, were recruited in this study. The baseline evaluation included physical examination, vital signs, laboratory tests, coagulation assessment, chest radiograph, and 12-lead electrocardiogram (12-ECG). The individuals with normal limits or clinically insignificant abnormalities of baseline evaluation were included. Exclusion criteria included known allergy to pleuromutilin antibiotics, lefamulin, or any of the excipients of the lefamulin formulations; history or symptoms of systemic disease; family history or presence of prolonged QTc syndrome, arrhythmia, syncope, or epilepsy (except for febrile convulsions in childhood); myocardial hypertrophy; other organic heart diseases, such as complete left bundle branch block; atrioventricular block; abnormal liver function; and breastfeeding females. All enrolled subjects provided written informed consent before participation.

### 4.3. Drug and Administration

The Lefamulin intravenous and oral formulations were produced by Patheon Italia S.p.A (batch no 00006) and Almac Pharma Services Limited (batch no W048475). The FDA and EMA release dates were 19 August 2019, and 27 July 2020, respectively. The structure formula of lefamulin is shown in Figure 6 The drug specifications were as follows: Tablets: 600 mg; IV infusion: 150 mg (15 mL), injected into a 250 mL citrate-saline buffer (pH = 5) for 60 (+5) min. The doses and administration methods of lefamulin in the two cohorts are presented in Table 6.

### 4.4. Blood Sample Collection, Processing, and Determination

In the single-dose group, blood samples were collected using K_3_EDTA tubes at various time points: 30 min before administration on day 1 (D1) in period 1 and at specific intervals after administration; 2 min (oral only), 10 min (oral only), 20 min (oral only), 30 min, 45 min (oral only), 1 h, 1.25 h, 1.5 h, 2 h, 2.5 h, 3 h, 4 h, 6 h, 9 h, 12 h, 15 h, 24 h, 36 h, and 48 h (before the first administration of multi -doses in period 2 at D3) after the administration; and within 30 min before the morning and evening administration at D7. In the multiple-dose group, blood samples were collected using K_3_EDTA tubes within 30 min before administration on D8 in Period 2, and at specific intervals thereafter: 10 min (oral only), 20 min (oral only), 30 min, 45 min (oral only), 1 h, 1.25 h, 1.5 h, 2 h, 2.5 h, 3 h, 4 h, 6 h, 9 h, 12 h, 15 h, 24 h, 36 h, and 48 h after the administration. Whole blood samples were centrifuged at 4 °C and 3000 rpm for 10 min. PK samples were centrifuged, separated, and frozen in an ultra-low temperature refrigerator (−70 °C ± 10 °C) within 1 h after collection for testing.

### 4.5. Determination of Blood Concentration

The concentration of lefamulin and its metabolite BC-8041 in PK plasma samples was determined by liquid chromatography-tandem mass spectrometry after protein precipitation treatment, with BC-3781-^13^C_3_, d2 and BC-3265 (both from Nabriva Therapeutics GmbH, Dublin, Ireland) as the internal standard. The concentration ranges of lefamulin and BC-8041 were 0.005–2.5 μg/mL and 0.001–0.5 μg/mL, respectively. The lower limit of quantification (LLOQ) was 0.005 μg/mL for lefamulin and 0.001 μg/mL for BC-8041.

### 4.6. PK Analysis

Non-compartmental analysis was performed using Phoenix WinNonlin (version 6.4 or above, Certara Co. Ltd., Princeton, NJ, USA) to calculate the PK parameters of lefamulin and its major metabolite BC-8041 based on the actual blood collection time points.

For single-dose administrations of lefamulin and its main metabolite BC-8041 in human plasma, the PK parameters included C_max_, T_max_, t_1/2_, AUC from 0 to last measurable concentration (AUC_0−t_), AUC_0−inf_, the AUC from 0 to 12 h (AUC_0–12 h_), the AUC from 0 to 24 h (AUC_0–24 h_), V_z_, V_z_ for oral administration V_z_ (V_z_/F), CL, and CL for oral administration (CL/F).

For multiple-dose administration of lefamulin and its main metabolite BC-8041 in human plasma, the PK parameters included C_max_ at steady state (C_max,ss_), minimum concentration at steady state (C_min,ss_), average concentration at steady state (C_avg,ss_), T_max_ at steady state (t_max,ss_), t_1/2, ss_, AUC_0–tau,ss_, AUC_0–24 h_ at steady state (AUC_0–24 h,ss_), V_z_ at steady state (V_z,ss_), V_z_/F at steady state (V_z,ss_/F), CL_ss_, CL/F at steady state (CL_ss_/F), accumulation ratio (Rac) based on AUC (R_ac(AUC)_), Rac based on C_max_ (R_ac(Cmax)_), and DF.

### 4.7. Susceptibility Study

The in vitro MIC distribution of lefamulin against clinically isolated 172 strains of *Streptococcus pneumoniae* and 121 strains of *Staphylococcus aureus* is illustrated in Table 7, based on the previously published susceptibility study data of lefamulin [5].

### 4.8. PK/PD Target

The *f*AUC/MIC was the PK/PD index that most strongly correlated with the efficacy of lefamulin [17]. Previous research [11] demonstrated that in the neutropenic murine pneumonia model, the *f*AUC/MIC (median) associated with 1-log_10_ cfu reduction from baseline was 1.37 for *Streptococcus pneumoniae* and 2.13 for *Staphylococcus aureus*.

### 4.9. PK/PD Analysis

Monte Carlo simulation was performed based on the steady-state exposure AUC_0–24,ss_ of lefamulin 150 mg IV infusion for 1 h. Different target values of *f*AUC/MIC were combined with the MIC distribution of lefamulin against *Streptococcus pneumoniae* and *Staphylococcus aureus* in vitro. According to previous studies [18], the protein binding rate of lefamulin varied with concentration, ranging from 74.1 to 97.4% at different concentrations after logistic regression. The free fraction (*f*) value was calculated according to the different protein binding rates (74.1%, 80%, 85%, 90%, 95%, and 97.4%). For each MIC value against both the isolates after 150 mg IV administration of lefamulin for 1 h, the PTA and the CFR were obtained. The effects of protein binding rate on PK/PD breakpoints and PTA were analyzed to evaluate the efficacy of the dosage regimen.

### 4.10. Safety Evaluation

Safety measures included adverse events (AE), physical examination, vital signs, laboratory tests, and 12-ECG.

## 5. Conclusions

In summary, lefamulin is safe and well-tolerated in healthy Chinese adults, both when administered intravenously and orally, with no new safety concerns identified. The PK characteristics of lefamulin in healthy Chinese subjects are similar to the results of previous studies conducted in other countries. It was microbiologically effective on *Streptococcus pneumoniae* and *Staphylococcus aureus* at the dose regimen in this study. However, it is important to note that this study was only a PK/PD analysis based on PK data from healthy subjects. Further research is required to obtain PK characteristics of lefamulin in CABP patients to evaluate the impact of disease status on PK and to conduct PK/PD analysis using the PK data of CABP patients. This will help in further evaluating the efficacy and safety of lefamulin in the treatment of pulmonary infection based on clinical usage and dosage.

## Figures and Tables

**Figure 1 antibiotics-12-01391-f001:**
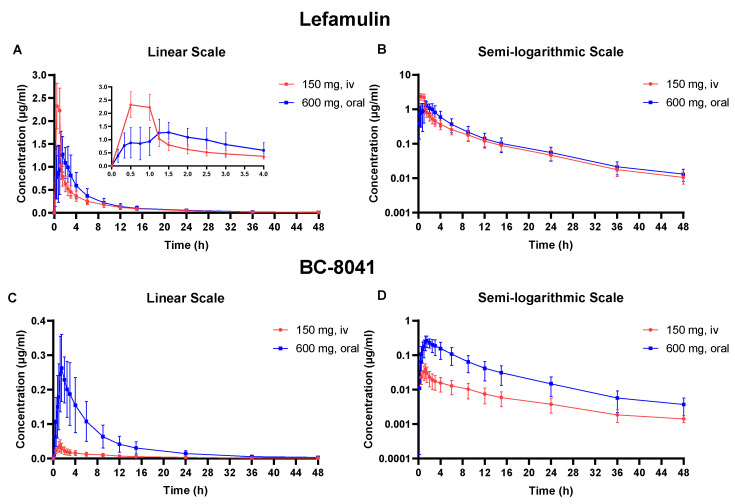
Mean plasma concentration-time curve of lefamulin and its metabolite BC-8041. (**A**) Linear scale graph of lefamulin after 150 mg intravenous (IV) or 600 mg oral administration; (**B**) semi-logarithmic graph of lefamulin after 150 mg IV or 600 mg oral administration; (**C**) linear scale graph of BC-8041 after 150 mg IV or 600 mg oral administration; (**D**) semi-logarithmic graph of BC-8041 after 150 mg IV or 600 mg oral administration.

**Figure 2 antibiotics-12-01391-f002:**
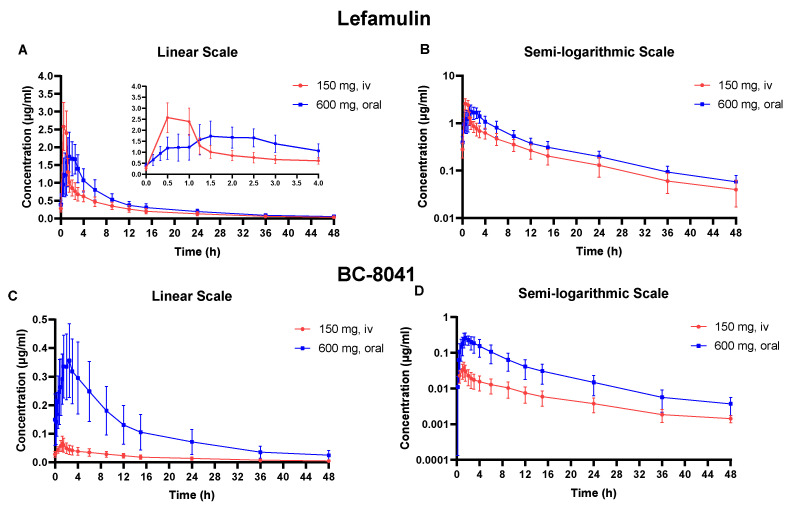
Mean plasma concentration-time curve of lefamulin and its metabolite BC-8041 at steady state. (**A**) Linear scale graph of lefamulin after 150 mg intravenous (IV) or 600 mg oral administration; (**B**) semi-logarithmic graph of lefamulin after 150 mg IV or 600 mg oral administration; (**C**) linear scale graph of BC-8041 after 150 mg IV or 600 mg oral administration; (**D**) semi-logarithmic graph of BC-8041 after 150 mg IV or 600 mg oral administration.

**Figure 3 antibiotics-12-01391-f003:**
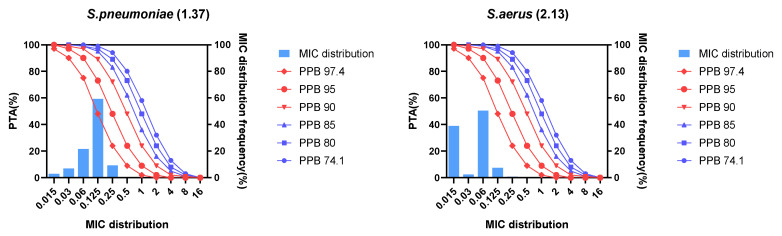
Probability of target attainment and MIC distribution for *Streptococcus pneumoniae* and *Staphylococcus aureus* after lefamulin 150 mg intravenous infusion 1 h. PPB, plasma protein binding rate; PTA, probability of target attainment; MIC, minimum inhibitory concentration; *S. pneumoniae*, *Streptococcus pneumoniae*; *S. aureus*, *Staphylococcus aureus*.

**Figure 4 antibiotics-12-01391-f004:**
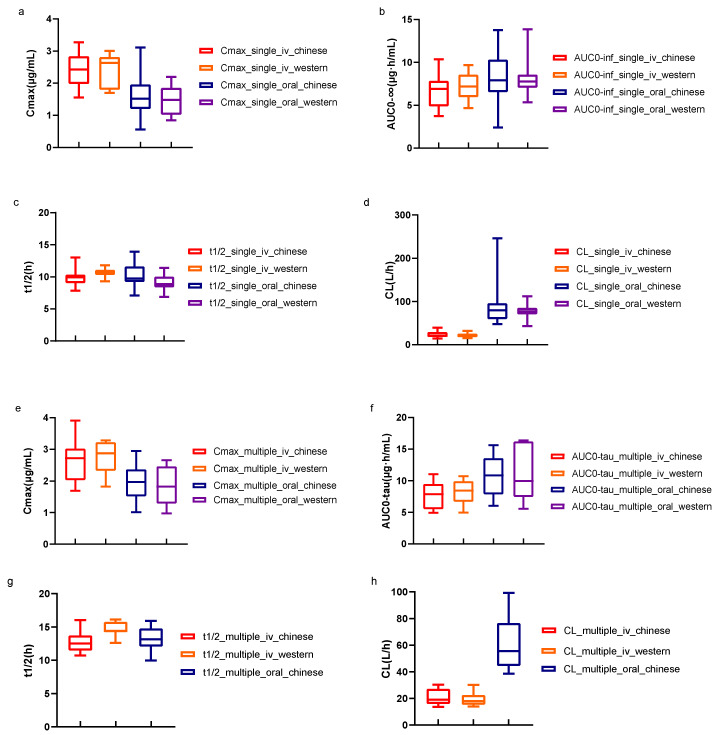
Comparison of PK parameters of lefamulin in healthy subjects between Chinese and Western populations. t_1/2_ and CL data of multiple-dose oral administration in other countries have not been released. The minimum, median, and maximum values for each parameter were displayed. iv, intravenous; C_max_, peak concentration; t_1/2_, elimination half-life; AUC_0−inf_, the area under concentration-time curve (AUC) from 0 to infinity; AUC_0−tau_, AUC within dosing interval; CL, apparent clearance. (**a***–***d**) shows the comparison of PK parameters of lefamulin after single administrations. (**e***–***h**) shows the comparison of PK parameters of Lefamulin after multiple administrations.

**Figure 5 antibiotics-12-01391-f005:**
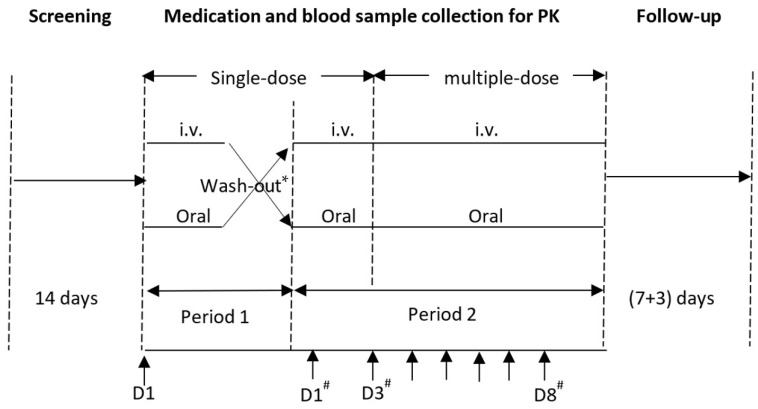
Study design and time axis. * The wash-out period between single-dose administration in period 1 and 2 was 7(+2) days. # Period 2; Medication.

**Figure 6 antibiotics-12-01391-f006:**
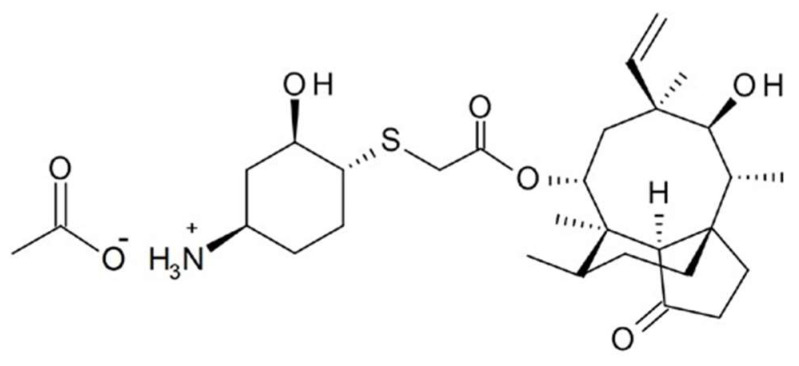
The structure formula of lefamulin.

**Table 1 antibiotics-12-01391-t001:** PK parameters of lefamulin in the plasma of healthy subjects after single or multiple doses of intravenous or oral administration.

Parameters	150 mg IV Administration (*N* = 20)	600 mg Oral Administration (*N* = 20) *	Parameters	150 mg IV Administration Every 12 h, D3–8 (*N* = 10)	600 mg Oral Administration Every 12 h, D3–8 (*N* = 10) *
C_max_ (μg/mL)	2.41 ± 0.51	1.56 ± 0.58	C_max,ss_ (μg/mL)	2.63 ± 0.67	1.94 ± 0.57
t_max_ (h)	NA	1.38, 0.35–3.00	C_min,ss_ (μg/mL)	0.27 ± 0.09	0.37 ± 0.11
t_1/2_ (h)	9.9 ± 1.2	10.15 ± 1.7	C_avg,ss_ (μg/mL)	0.65 ±0.17	0.89 ± 0.27
AUC_0–t_ (μg·h/mL)	6.75 ±1.79	7.51 ± 2.76	t_max,ss_ (h)	NA	1.75, 1.25–3.00
AUC_0–inf_ (μg·h/mL)	6.90 ± 1.84	7.71 ± 2.82	t_1/2,ss_ (h)	12.7 ± 1.6	13.2 ± 1.9
AUC_0–12 h_ (μg·h/mL)	5.34 ± 1.34	5.87 ± 2.08	AUC_0–24 h,ss_ (μg·h/mL)	10.03 ± 2.87	13.91 ± 4.10
AUC_0–24 h_ (μg·h/mL)	6.24 ± 1.61	6.91 ± 2.51	AUC_0−tau,ss_ (μg·h/mL)	7.84 ± 2.07	10.64 ± 3.23
V_z_ (L)	332.7 ± 100.0	1308 ± 542.6	V_z,ss_ (L)	368.6 ± 90.2	1177 ± 412.4
CL (L/h)	23.5 ±7.2	92.5 ± 48.0	CL_ss_ (L/h)	20.5 ± 5.9	61.6 ± 20.0
/	/	/	R_ac(Cmax)_	1.0 ± 0.1	1.3 ± 0.1
/	/	/	R_ac(AUC)_	1.4 ± 0.1	1.6 ± 0.3
/	/	/	DF (%)	363.6 ± 32.8	178.4 ± 27.9

Data are presented as geometric mean ± standard deviation or median, minimum–maximum unless otherwise specified. IV, intravenous; C_max_, peak concentration; C_min_, minimum concentration; C_avg_, average concentration; t_max_, time to peak concentration; t_1/2_, elimination half-life; AUC_0−t_, the area under concentration-time curve (AUC) from 0 to last measurable concentration; AUC_0−inf_, AUC from 0 to infinity; AUC_0–12 h_, AUC from 0 to 12 h; AUC_0–24 h_, AUC from 0 to 24 h; AUC_0−tau_, AUC within dosing interval; V_z_, apparent volume of distribution; CL, apparent clearance; _SS_, steady state; R_ac(Cmax)_, accumulation ratio (Rac) based on C_max_; R_ac(AUC)_, Rac based on AUC; DF, degree fluctuation. * The corresponding PK parameters for oral administration are V_z_/F, CL/F, and F was bioavailability.

**Table 2 antibiotics-12-01391-t002:** PK parameters of the main metabolite of lefamulin, BC-8041, in the plasma of healthy subjects after single or multiple doses of intravenous or oral administration.

Parameters	150 mg IV Administration (*N* = 20)	600 mg Oral Administration (*N* = 20) *	Parameters	150 mg IV Administration Every 12 h, D3–8 (*N* = 10)	600 mg Oral Administration Every 12 h, D3–8 (*N* = 10) *
C_max_ (μg/mL)	0.04 ± 0.01	0.3 ± 0.10	C_max,ss_ (μg/mL)	0.07 ± 0.03	0.38 ±0.12
t_max_ (h)	NA	1.50, 0.75–3.00	C_min,ss_ (μg/mL)	0.02 ± 0.01	0.13 ± 0.07
t_1/2_ (h)	11.59 ±2.89	9.72 ± 1.43	C_avg,ss_ (μg/mL)	0.04 ± 0.01	0.23 ± 0.10
AUC_0−t_ (μg·h/mL)	0.28 ± 0.13	1.84 ± 0.82	t_max,ss_ (h)	NA	2.25, 1.25–2.50
AUC_0−inf_ (μg·h/mL)	0.30 ± 0.14	1.90 ± 0.85	t_1/2,ss_ (h)	14.57 ± 2.23	14.71 ± 2.34
AUC_0–12 h_ (μg·h/mL)	0.18 ± 0.08	1.37 ±0.56	AUC_0–24 h,ss_ (μg·h/mL)	0.63 ± 0.23	4.0 ± 1.80
AUC_0–24 h_ (μg·h/mL)	0.24 ± 0.11	1.68 ±0.73	AUC_0−tau,ss_ (μg·h/mL)	0.43 ± 0.15	2.81 ± 1.15
V_z_ (L)	NA	NA	V_z,ss_ (L)	NA	NA
CL (L/h)	NA	NA	CL_ss_ (L/h)	NA	NA
/	/	/	R_ac(Cmax)_	1.80 ± 0.42	1.14 ± 0.21
/	/	/	R_ac(AUC)_	2.64 ± 0.62	1.66 ± 0.36
/	/	/	DF (%)	120.0 ± 21.8	116.6 ± 29.5

Data are presented as geometric mean ± standard deviation or median, minimum–maximum unless otherwise specified. IV, intravenous; C_max_, peak concentration; C_min_, minimum concentration; C_avg_, average concentration; t_max_, time to peak concentration; t_1/2_, elimination half-life; AUC_0−t_, the area under concentration-time curve (AUC) from 0 to last measurable concentration; AUC_0−inf_, AUC from 0 to infinity; AUC_0–12 h_, AUC from 0 to 12 h; AUC_0–24 h_, AUC from 0 to 24 h; AUC_0−tau_, AUC within dosing interval; V_z_, apparent volume of distribution; CL, apparent clearance; _SS_, steady state; R_ac(Cmax)_, accumulation ratio (Rac) based on C_max_; R_ac(AUC)_, Rac based on AUC; DF, degree fluctuation. * The corresponding PK parameters for oral administration are V_z_/F, CL/F, and F was bioavailability.

**Table 3 antibiotics-12-01391-t003:** Probability of target attainment for *Streptococcus pneumoniae* (**a**) and *Staphylococcus aureus* (**b**) after lefamulin 150 mg intravenous infusion for 1 h.

(**a**)							
***S. pneumoniae* (MIC_90_ = 0.125 mg/L)**							
PK/PD Target		1.37 (1-log_10_ cfu Reduction)					
Protein Binding Rate		74.1	80	85	90	95	97.4
MIC (mg/L)	0.015	100	100	100	100	99	97
	0.03	100	99	99	99	97	94
	0.06	99	99	98	97	94	88
	0.125	98	97	95	93	86	73
	0.25	95	94	90	86	73	55
	0.5	89	86	81	72	54	36
(**b**)							
** *S. aureus* ** **(MIC_90_ = 0.06 mg/L)**							
PK/PD Target		2.13 (1-log_10_ cfu Reduction)					
Protein Binding Rate		74.1	80	85	90	95	97.4
MIC (mg/L)	0.015	100	100	100	99	98	95
	0.03	99	99	99	97	95	90
	0.06	98	98	97	95	89	80
	0.125	96	94	93	89	78	63
	0.25	91	89	85	77	60	43
	0.5	83	78	71	60	41	25

*S. pneumoniae*, *Streptococcus pneumoniae*; *S. aureus*, *Staphylococcus aureus*.

**Table 4 antibiotics-12-01391-t004:** Cumulative fraction of response for *Streptococcus pneumoniae* and *Staphylococcus aureus* after lefamulin 150 mg intravenous infusion for 1 h.

Protein Binding Rate	Target	*S. pneumoniae* (MIC_90_ = 0.125 mg/L)	Target	*S. aureus* (MIC_90_ = 0.06 mg/L)
74.1	1.37	98	2.13	99
80		97		98
85		97		98
90		94		96
95		91		92
97.4		88		86

*S. pneumoniae*, *Streptococcus pneumoniae*; *S. aureus*, *Staphylococcus aureus*.

**Table 5 antibiotics-12-01391-t005:** Summary of Treatment-related TEAEs by System Organ Class and Preferred Term.

	150 mg IV Administration (*N* = 20), *n* (%)	600 mg PO Administration (*N* = 20), *n* (%)
Treatment-related TEAEs	11 (55.0)	12 (60.0)
Gastrointestinal disorders	7 (35.0)	13 (65.0)
Nausea	2 (10.0)	5 (25.0)
Abdominal discomfort	1 (5.0)	4 (20.0)
Abdominal pain upper	1 (5.0)	3 (15.0)
Abdominal pain	0	3 (15.0)
Diarrhoea	0	3 (15.0)
General disorders and administration site conditions	10 (50.0)	0
Infusion site pain	9 (45.0)	0
Infusion site pruritus	8 (40.0)	0
Infusion site erythema	5 (25.0)	0
Infusion site swelling	5 (25.0)	0
Infusion site induration	4 (20.0)	0
Infusion site haemorrhage	1 (5.0)	0
Investigations	7 (35.0)	4 (20.0)
Blood creatinine increased	2 (10.0)	0
Eosinophil percentage increased	1 (5.0)	1 (5.0)
Blood creatine phosphokinase increased	0	1 (5.0)
Eosinophil count increased	1 (5.0)	0
Nervous system disorders	3	4 (20.0)
Headache	1	3 (15.0)
Dizziness	0	1 (5.0)
Ear and labyrinth disorders	0	1 (5.0)
Ear pain	0	1 (5.0)
Respiratory, thoracic and mediastinal disorders	0	1 (5.0)
Oropharyngeal pain	0	1 (5.0)

**Table 6 antibiotics-12-01391-t006:** Doses and administration methods of lefamulin in the two cohorts.

Cohort	Period 1 Single-Dose, D1	Wash-Out Period (Days)	Period 2	
			Single-Dose, D1	Multiple-Dose, D3−8 (Administration Only in the Morning on D8)
1	150 mg IV administration	7	600 mg oral administration	600 mg oral administration, every 12 h
2	600 mg oral administration	7	150 mg IV administration	150 mg IV administration, every 12 h

**Table 7 antibiotics-12-01391-t007:** MIC distribution of lefamulin against *Streptococcus pneumoniae* and *Staphylococcus aureus*.

Bacteria (No. of Isolates)	Frequency Distribution (%) of MIC (mg/L)					MIC_50_/MIC_90_
	0.015	0.03	0.06	0.125	0.25	
*Streptococcus pneumoniae* (172)	2.91	6.98	21.51	59.3	9.3	0.125/0.125
Cumulative frequency distribution of *Streptococcus pneumoniae*	2.91	9.88	31.40	90.70	100.00	\
*Staphylococcus aureus* (121)	38.84	2.48	50.41	7.44	0.83	0.06/0.06
Cumulative frequency distribution of *Staphylococcus aureus*	38.84	41.32	91.74	99.17	100.00	\

## Data Availability

Not applicable.

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
