# Peer review of "Pharmacokinetic, Pharmacokinetic/Pharmacodynamic, and Safety Investigations of Lefamulin in Healthy Chinese Subjects"

_antibiotics, 2023, doi:10.3390/antibiotics12091391_

Round 1

Reviewer 1 Report

Comments see attached file.

Author Response

  1. Line 45: Reference [2] refers to a Canadian surveillance study. Please consider including references to global surveillance studies that have been published in journals or at various scientific conferences. Authors may want to visit the webpage of Nabriva for consultation (https://www.nabriva.com/news).

Response: We have cited the surveillance reports from SENTRY as our references. (ref 3 & ref 4)

  1. Line 48: Authors should provide references for the statement that the in vitro activity of lefamulin is similar in for Chines clinical isolates and isolates from Europe and America.

Response: We have supplemented the in vitro activity data of lefamulin from SENTRY in 2015-2017 to verify that the in vitro activity of lefamulin is similar for Chinese clinical isolates and isolates from Europe and America. (ref 3 & ref 4)

  1. Line 58: Please consider including the references to the respective webpages of FDA and EMA rather than citing a review article [6] only.

Response: We have included the references to the respective webpages of FDA and EMA. (ref 9 and ref 10)

  1. Lines 142-160, Section 2.3 PK/PD analysis and Lines 238-247 of corresponding Discussion Section: Authors performed a PTA analysis of lefamulin considering various protein binding rates, but it remains unclear in the Results section and further on in the Discussion section, why this was done and why a plasma protein binding range of 74.1-97.4% was considered. The authors only cite two reports (references [13] and [14]), which are not publicly available. In my opinion, authors must only refer to published literature, either journal publications or publicly available poster presentations. According to my knowledge, plasma protein binding data and free drug concentrations of 12-27%, respectively, have been published by Wicha W. et al, JAC 2019; 74 Suppl 3: iii19-iii26. Furthermore, in-vitro plasma protein binding of 73-88% in humans have been published in the EMA assessment report EMA/325848/2020 (http://www.ema.europe.eu/en/documents/assessment-report/xenleta-epar-public-assessment-report en.pdf). Therefore, it appears that a plasma protein binding range of 73-88% is more relevant than the range used by the authors. If otherwise, authors should explain the reasons for the broad range of protein binding and potentially discuss the methods used determination. Moreover, authors should provide insight into that the binding of lefamulin to plasma proteins is very weak or loose and that MIC values are hardly affected by the presence of plasma proteins (see also in EMA assessment report EMA/325848/2020). Authors may also want to discuss the limited relevance of plasma protein binding data for the ELF concentrations.

Response: Regarding the protein binding rate, we referred to the FDA Multi-discipline review of Lefamulin (Page 211, Table 102, link: https://www.accessdata.fda.gov/drugsatfda_docs/nda/2019/211672Orig1s000,%20211673Orig1s000MultidisciplineR.pdf).

According to the review, lefamulin displayed different plasma protein binding rates of 86-97% and 73-88% in study XS-1103 and EVT-00756-3781, respectively. FDA reviewers considered the plasma protein binding of lefamulin (73% to 88%) appears to be underestimated in study EVT-00756-3781 since plasma protein binding was determined using pooled blank plasma diluted to 85% (v/v) following the addition of lefamulin solution. Thus, we chose a range of plasma protein binding rates to conduct the PK/PD analysis. The maximum number of 97.4% and the minimum number of 74.1% were acquired from model fitting using the PPB data from the study XS-1103 and EVT-00756-3781 in the FDA report. The model was shown as follows (Page 236 in FDA Multi-discipline review), Where Fu is the fraction unbound of lefamulin in plasma with minimum value of Fumin and maximum value of Fumax. Fu50 is the concentration at 50% Fumax.

In EMA assessment report, it was mentioned that the protein binding rates of 73-88% conflicted with those of normal control population (94.8-97.1%) in the renal impairment study and hepatic impairment study (Page 87). Considering all the above points, we prefer to use the protein binding rates ranging from 74.1 to 97.4%.

We acknowledged that free ELF concentrations should be acquired to further evaluate the lefamulin efficacy to CAP. We added this point in lines 250-252.

  1. Lines 142-178, Section 2.3 PK/PD analysis and Figures 3-1 and 3-2 and Table 4: Authors performed PK/PD target attainment analysis for pneumoniae and for S. aureus up to a lefamulin MIC of 0.25 μg/mL. Since the lefamulin susceptible breakpoint is 0.5 μg/mL for S. pneumoniae, it would be important to present the PTA also for the lefamulin MIC of 0.5 μg/mL. Secondly, my recommendation would be to exclude the PTA for the protein binding rates above 88% in order to avoid any confusion of published target attainment data and breakpoint justifications and because the broad protein binding range is rather caused by the methodology used for plasma protein binding determination rather than by inter-subject variations within a population.

Response: We have supplemented the PTA results for Lefamulin MIC of 0.5 μg/mL. (Table 3-1 and Table 3-2). In Figure 3, the MIC distribution and PTA at MIC level of 0.015-16 μg/mL have been included in our first submission. The issue of protein binding rates has been explained in point 5, thus we prefer to keep the present results using protein binding rates ranging from 74.1% to 97.4%.

  1. Lines 164, 172, 173, 197: Please fix the spelling errors in the species names of the in the Figure legends (S. pneumoniae, Streptococcus pneumoniae, Chlamydophila pneumoniae).

Response: The spelling errors have been corrected. (Lines 164, 172, 173, 197)

  1. Lines 218-219: Authors state that half of the subjects who received oral administration of lefamulin showed a bimodal curve, with the first peak occurring 20 to 45 minutes after administration and the second, higher peak occurring 1-2 h after administration. This statement cannot be verified from the Figures provided in the manuscript.

Response: We have added partially enlarged details near the main figure. (Figure 1A & Figure 2A)

  1. Figure 4: Authors should revise the Legend of the graphs, since currently the reader has to guess if “Ch" means Chinese; unclear what “en" is the abbreviation of. Most importantly, authors should provide references to the publication for the PK data of the non-Chinese populations. Please note that the supplement JAC 2019; 74 Suppl 3 also provides PK data for multiple oral dosing regimes that could be considered for Figures 4g-h.

Response: We have changed the abbreviation to the full name to avoid misunderstanding. The meanings of all the abbreviations in Fig. 4 were supplemented. We also added the reference as ref 15 in the revised version as you suggested.

  1. Figures 1-2 and associated Results/Discussion sections: authors should provide insight about the relevance of the PK data for the metabolite BC-8041 and describe and discuss the data in the relevant sections.

Response: The PK characteristics of the metabolite BC-8041 have been displayed in Section 2.2 (Lines 76-98). We added the discussion about the relationship of lefamulin and metabolite BC-8041 as follows: Lefamulin is primarily metabolized by CYP3A4 and the metabolite BC-8041 is the main component that displayed similar elimination with lefamulin. The metabolite rates were estimated to be 4% and 25% for intravenous and oral formulation, respectively. (Lines 227-230)

  1. Lines 279-282, Drug and administration: Authors should provide the sources of lefamulin tablets and IV formulations.

Response: The Lefamulin intravenous and oral formulations were produced by Patheon Italia S.p.A (batch no 00006) and Almac Pharma Services Limited (batch no W048475). We have supplemented the source and batch number of the investigational drugs in Section 4.3 Drug and Administration.

  1. Lines 284, Blood sample collection: Authors should specify which tubes were used for plasma collection.

Response: K3EDTA tubes were used for blood sampling. (Lines 299, 305)

  1. Line 300: BC-3781-13C3 was used as the internal standard. Authors should provide insight about the source of this standard and may want to explain that BC-3781 is actually the code for lefamulin.

Response: We added the internal standard for metabolite BC-8041 and the sources of these two internal standards were both from Nabriva Therapeutics GmbH. (Lines 315-316)

  1. Line 323, Table 7: Typically, the cumulative frequency distribution (%) by MIC is provided in literature rather than the just the frequency by MIC. Authors should revise the table and extend the by additional columns for MICs covering the full range of MICs (may be 0.5 μg/mL or higher).

Response: We have added the cumulative frequency distribution in Table 7 and MIC of 0.25 is the highest level for S. pneumoniae and S. aureus.

  1. Funding: If the study was supported by a pharma company in addition to the mentioned research grant, this should be stated. It is hard to believe that this study was the initiative of the Huashan Hospital in Shanghai only.

Response: We have claimed that the phase I clinical study was sponsored and funded by Sinovant Sciences in the funding section. (Line 377)

  1. Lines 409-413, References: Authors shouldn't refer to study reports that aren't publicly available but should rather cite manuscripts published in peer-reviewed journals.

Response: We have changed the reference to the FDA Multi-Disciplinary Review and Evaluation that can be acquired on the FDA website.

Reviewer 2 Report

I think that the manuscript can be accepted in this form.

I have no objections to the text.

1. What is the main question addressed by the research? This study aimed to explore the pharmacokinetics (PK) and safety of oral (PO) and intravenous (IV) lefamulin in healthy Chinese subjects and to evaluate the efficacy of the intravenous administration regimen by pharmacokinetics/pharmacodynamic (PK/PD) analysis.   2.  Do you consider the topic original or relevant in the field, and if so, why? Lefamulin is the first pleuromutilin antibiotic used for the systemic treatment of bacterial infections in humans. Lefamulin was approved by the United States Food and Drug Administration (FDA) in 2019 and the European Medicines Agency (EMA) in 2020 for the treatment of CABP based on the completed clinical studies. However, the pharmacokinetics (PK) and safety of lefamulin have not been investigated in healthy Chinese participants (both oral and IV infusion). It is important to conduct pharmacokinetic/pharmacodynamic (PK/PD) analysis based on the PK of lefamulin in healthy Chinese subjects and the distribution of minimum inhibitory concentration (MIC) of common pathogenic bacteria causing CABP. This analysis is crucial for the rational use of lefamulin in Chinese patients with CABP.   3. What does it add to the subject area compared with other published material? This is the first study evaluating the safety and PK profile of lefamulin following intravenous and oral administrations at single and multiple doses in healthy Chinese subjects and support clinical application of lefamulin in Chinese population from PK/PD perspective.   4. What specific improvements could the authors consider regarding the methodology? The PK/PD analysis was performed properly. Regarding the safety statement, N=20 is rather low to declare this a "safety" study as well.   5. Are the conclusions consistent with the evidence and arguments presented and do they address the main question posed? Yes, the conclusions are consistent and properly address the main questions posed.    6. Are the references appropriate? Yes, the references are appropriate.   7. Please include any additional comments on the tables and figures. Figure 4: - The caption should state the parameters in the Box and Whisker plot (mean/median, SD/90CI,,,).   - The legends take up too much space compared to the plots on panels a-h.

Author Response

  1. Please include any additional comments on the tables and figures. Figure 4: - The caption should state the parameters in the Box and Whisker plot (mean/median, SD/90CI,,,).   - The legends take up too much space compared to the plots on panels a-h.

Response: We have stated the parameters in the Box and Whisker plot in Figure 4. They represent minimum, median and maximum values.

Reviewer 3 Report

Lefamulin is a semi-synthetic antibiotic of the pleuromutilin class. Indications for the use of the drug is the treatment of community-acquired pneumonia caused by the following microorganisms: Streptococcus pneumoniae, Staphylococcus aureus, Haemophilus influenzae, Legionella pneumophila, Mycoplasma pneumoniae, Chlamydophila pneumoniae.

The authors conducted a clinical study of Lefamulin - this study was aimed at studying the pharmacokinetics and safety of oral and intravenous lefamulin in healthy Chinese subjects, as well as evaluating the effectiveness of the intravenous regimen using pharmacokinetic / pharmacodynamic analysis. Undoubtedly, the practice of introducing medicines approved for use in other countries requires additional study of these drugs, especially when it comes to patients living in other regions and having a different ethnicity. In this regard, in my opinion, this study has practical significance and could potentially be accepted for publication in the journal Antibiotics.

In the meantime, I have a number of comments and suggestions:

1. It would not be superfluous to give the structural formula of the compound under study.

2. I did not find any data on the origin of lefamulin used for clinical trials. Did the team of authors use ready-made forms of the drug or was the active substance synthesized? If the former, is it possible to indicate a specific manufacturer, series and release date? If the latter, by what method and what is the purity of the Lefamulin?

3. How representative is the sample of 20 patients, which is also disaggregated by age, gender, and also divided into patients receiving the drug orally and as an injection?

4. Why were Streptococcus pneumoniae and Staphylococcus aureus chosen for the study?

Author Response

  1. It would not be superfluous to give the structural formula of the compound under study.

Response: The structural formula of Lefamulin has been added in Figure 5.

  1. I did not find any data on the origin of lefamulin used for clinical trials. Did the team of authors use ready-made forms of the drug or was the active substance synthesized? If the former, is it possible to indicate a specific manufacturer, series and release date? If the latter, by what method and what is the purity of the Lefamulin?

Response: We used the intravenous and oral formulations of lefamulin produced by Patheon Italia S.p.A (batch no 00006) and Almac Pharma Services Limited (batch no W048475). The FDA and EMA release dates were August 19th, 2019, and July 27th, 2020, respectively.

  1. How representative is the sample of 20 patients, which is also disaggregated by age, gender, and also divided into patients receiving the drug orally and as an injection?

Response: We recruited 20 healthy subjects rather than patients in Phase I clinical trial. They were representative of the general healthy population. Each of these 20 subjects received both intravenous and oral formulations of lefamulin. As a result, we were able to calculate the PK parameters for a single dose based on the drug concentrations observed in these 20 subjects.

  1. Why were Streptococcus pneumoniae and Staphylococcus aureus chosen for the study?

Response: Streptococcus pneumoniae is the most common pathogen of community acquired pneumonia, which is responsible for almost 50% of cases. Staphylococcus aureus, specifically, methicillin-resistant S. aureus (MRSA) can cause severe pneumonia that leads to critical illness and death in CAP patients. Thus, we chose these two species in this study.